# KIMMDY: a biomolecular reaction emulator

Eric Hartmann [1,2,6], Jannik Buhr [1,2,6], Kai Riedmiller [1,2,6], Evgeni Ulanov[1,3,4], Boris N. Schüpp[3,5], Denis Kiesewetter [2,3], Daniel Sucerquia [1,3,4], Camilo Aponte-Santamaría [3] & Frauke Gräter [1,2,3,5] ✉

Molecular simulations have become indispensable in biological research. Their accuracy continues to improve, but directly modelling biochemical reactions – central to all life processes – remains computationally challenging. Here, we present a biomolecular reaction emulator that models reactions across conformational ensembles using kinetic Monte Carlo. Our method, KIMMDY, is capable of handling dynamic, large-scale systems with successive, competing reactions, even on the second timescale or slower. It leverages graph neural networks for large-scale prediction of reaction rates, while also being capable of using simpler physics-based or heuristic models. We validate our approach against experimental data and showcase its power and versatility through a series of applications, including radical reactions, nucleophilic substitutions, and photodimerization. Example systems span proteins and DNA. KIMMDY aids the understanding of biochemical reaction cascades in complex systems, helps to re-interpret experimental data, and can inspire future wet-lab experiments.

Biomolecules are inherently reactive. Their reactivity enables essential biological processes. Nature makes use of a plethora of ingenious biochemical reactions, from nucleophilic substitutions and additions to rearrangements and redox reactions, to drive essential biological processes, including metabolism, signalling, and energy transfer. Beyond these tightly controlled and specific reactions, biomolecules undergo a wide range of unspecific reactions during their lifespan while they are subjected to varying conditions, such as pH, light or oxidative stress, causing constant molecular damage under physiological conditions.

These biochemical processes happen within and are driven by the complex, crowded and dynamic molecular environment inside and outside cells. Molecular simulations of such systems have given unprecedented insights into the inner workings of cells, thereby helping to overcome the fundamental limitation in the spatio-temporal resolution of experiments. Length scales and molecular complexities of the atomistic or coarse-grained models in biomolecular simulations are approaching, or already reaching, those of whole cells[1,2]. However, these simulations lack the key feature of life: reactivity, which has to be sacrificed for efficiency. Molecular mechanics (MM) force fields are the prime choice to handle large systems at sufficient timescales, but bonds cannot break and form in these models. Monitoring the interplay of molecular motion and chemical events, however, is key to understanding and designing complex molecular systems.

To overcome the limitations of classical simulations, reactive force fields[3,4], or schemes that allow for specific reactions to occur, such as the empirical valence bond method[5], or constant pH simulations[6] have been put forward. Also, the more recent machine-learned interatomic potentials (MLIPs), currently transforming the role of atomistic modelling, are by design reactive[7–12].

All of these models—MLIPs, classical reactive, and even the highly efficient unreactive force fields—face the fundamental limitation that the reachable timescales are far below those of most biochemical reactions. The same naturally applies to hybrid approaches such as quantum mechanics/molecular mechanics (QM/MM) and machine learning/molecular mechanics (ML/MM), whose performance is kept back by the slower method of the two (Fig. 1 top). Even if such hybrid

[1]Heidelberg Institute for Theoretical Studies, Am Schloss-Wolfsbrunnenweg 35, Heidelberg, Germany. [2]Institute for Scientific Computing, Heidelberg University, Im Neuenheimer Feld 205, Heidelberg, Germany. [3]Max Planck Institute for Polymer Research, Ackermannweg 10, Mainz, Germany. [4]Department of Physics and Astronomy, University of Heidelberg, Im Neuenheimer Feld 226, Heidelberg, Germany. [5]Max Planck School Matter to Life, Jahnstraße 29, Heidelberg, Germany. [6]These authors contributed equally: Eric Hartmann, Jannik Buhr, Kai Riedmiller. ✉e-mail: graeter@mpip-mainz.mpg.de

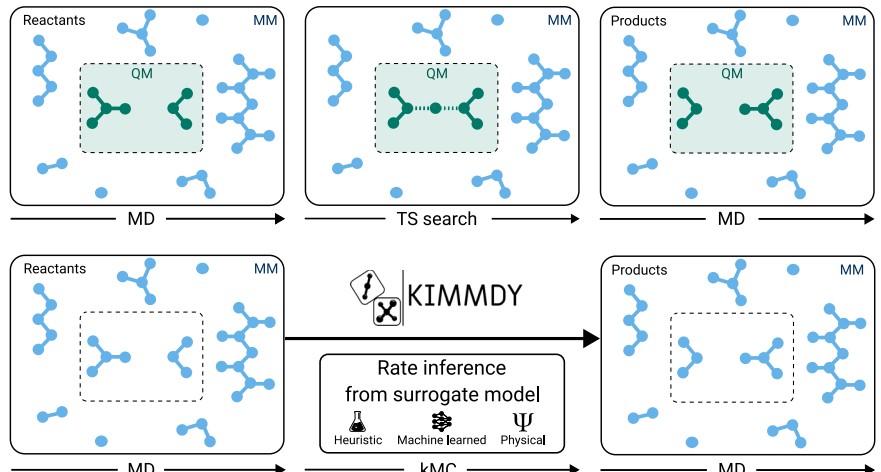

**Fig. 1 | KIMMDY is opposed to a QM/MM scheme.** Comparison between simulating reactions in QM/MM (or related schemes such as in MLIP or MLIP/MM, top) and emulating them with KIMMDY (bottom). Instead of performing costly transition state search with QM (or MLIP) methods, KIMMDY extracts reactant configurations from the MD system and performs reactions via the kMC step, by inferring rates from a reaction-specific model (more details in Fig. 2).

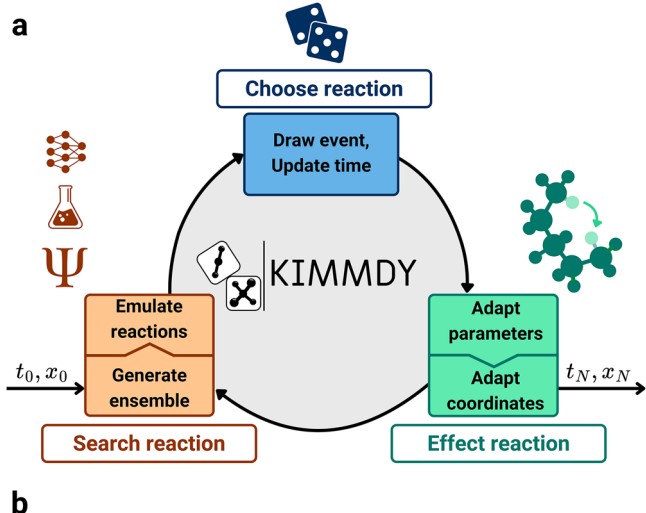

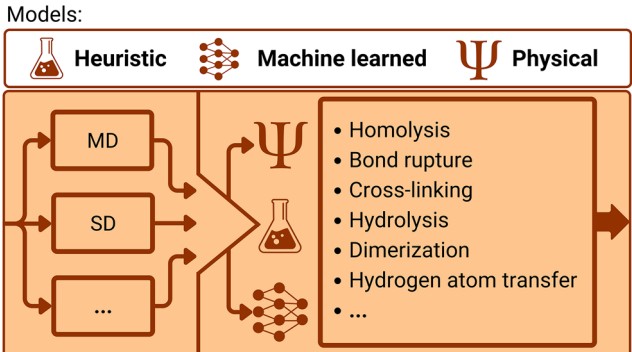

**Fig. 2 | KIMMDY as a biomolecular reaction emulator. a** Visualisation of the central workflow. KIMMDY searches for reactions within a dynamic molecular system, predicts reaction rates based on a learned, heuristic (e.g. experiment-based) or physical model, chooses one of those reactions by kMC, and then effects the reaction. **b** Detailed scheme of the Search reaction step of KIMMDY, displaying possible input ensemble generators and reaction types. MD and SD refer to molecular and stochastic dynamics.

approaches become more efficient in the future, they are unfeasible when it comes to the rich chemistry of biomolecules, being it post-translational modifications or radical chain reactions, where reactive regions can essentially span the whole system and timescales of reactions can reach seconds. Overcoming these limitations would enable large-scale simulations of reactive biomolecular systems.

We here present KIMMDY (KInetic Monte carlo Molecular DYnamics), a biomolecular reaction emulator (Fig. 1, bottom). KIMMDY searches for reactions within a dynamic molecular system and predicts reaction rates based on an emulator, that is, a machine-learned, heuristic or physical surrogate model (Fig. 2a). KIMMDY subsequently chooses one of those reactions in a kinetic Monte Carlo (kMC) step, effects this reaction and generates a new conformational ensemble to emulate further reactions. While combinations of kMC and MD have been proposed before[13] and also while this work was under review[14], we here—to our knowledge for the first time—achieve a direct dependence of the chemical kinetics, i.e. reaction rates, on the conformational dynamics of the reactants. KIMMDY can predict reaction dynamics in highly reactive and large biomolecular systems, can cope with reaction cascades therein as well as competing reactions, and can, in principle, handle reactions at virtually any timescale.

While combinations of MD and kMC have been used previously[15–17], our approach directly takes conformational ensembles into consideration and harnesses the power of graph neural networks (GNN) to rapidly predict ensemble-based reaction barriers[18]. At the same time, we profit from the validated accuracy and speed of classical biomolecular physics-based force fields[19,20] and the ease of extending them to new chemistries by ML[21]. We validate KIMMDY against experiments and demonstrate its applicability to open and closed-shell reactions and protein and nucleic acid systems (Fig. 2b). KIMMDY is implemented as a modular platform that can be extended to arbitrary reactions, providing reaction rates and force fields are accessible via ML-models or through experimental data. To properly demonstrate this extensibility, this work includes results obtained from a wide range of reactions and systems.

## Results

### Emulated reaction dynamics correctly predict radical reactions in small molecules

As a first area of application, we chose hydrogen atom transfer (HAT). HAT is ubiquitous in biomolecules and soft matter, can occur

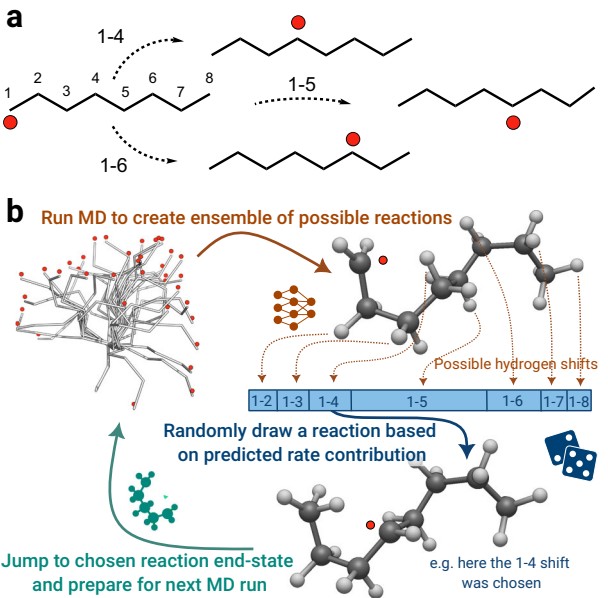

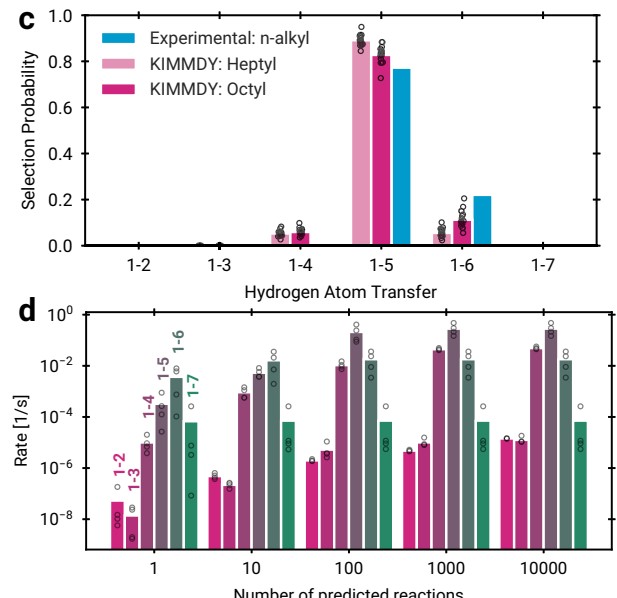

**Fig. 3 | Validating KIMMDY: Hydrogen atom transfer (HAT) in *n*-alkyl radicals.**
**a** Illustration of 1–4, 1–5 and 1–6 HATs in a 1-octyl radical. Unpaired electrons are shown as red dots. **b** KIMMDY HAT prediction cycle. KIMMDY searches for possible HAT reactions within an MD-generated conformational ensemble, infers their barriers from an emulator, chooses one reaction by kMC, adjusts the topology, and a new MD simulation commences. **c** Probability of HAT reactions as derived with KIMMDY for heptyl and octyl radicals, and compared with the experimental probability of *n*-alkyl. Experimental rates for a given shift were averaged over all

*n*-alkyl radicals where available (see Supplementary Methods). Circles indicate a single 100 ns KIMMDY run (20 independent runs) and coloured bars signify the mean. For shifts that were not available experimentally, i.e. 1–2 and 1–7, the rates are shown in Fig. S2, but the probabilities are set to 0 in (**c**). Hydrogen shifts between primary carbons cannot be detected in experiments and are excluded here. **d**, Mean rates for different numbers of barrier predictions per reaction (4 octyl runs, 10 ns, 500 K). Further details in Supplementary Methods. Source data are provided as a Source Data file.

unspecifically and in cascades of many subsequent radical migration steps, often on timescales of seconds, and is thus a reaction type where standard approaches such as QM/MM or direct MD simulations with machine-learned potentials fail. We started with validating KIMMDY for HAT within small hydrocarbon radicals, for which experimental results are available. The systems consisted of *n*-alkane-derived radicals, starting with the 1-propyl radical up to the 1-octyl radical (Fig. 3a). The machine-learned barrier prediction model used to infer rates for kMC was trained on barriers calculated at the DFT level for HAT reactions for proteins[18], which we hypothesised to sufficiently generalise to hydrocarbons (see Supplementary Methods). We would like to emphasise that we here infer rates from reaction barriers predicted solely from reactant structures, as opposed to performing a costly transition state search using QM or an MLIP.

Figure 3b shows the typical KIMMDY workflow, exemplified with an octyl radical. Figure 3c shows the selection probability for different HAT reactions at 500 K together with experimental results. We chose the mean of the experimentally-derived rates as a reference for a given hydrogen shift divided by the sum of all mean experimentally derived rates of all considered shifts. Similarly, we extracted the selection probability for these shifts from KIMMDY simulations. In line with the experimental results, our method correctly identifies 1–5 as the dominant HAT, due to a favourable six-ring configuration of the transition state, and also accurately predicts the overall trend of the 1−*n* shift probabilities, particularly, the low probabilities of transitions with high ring strain energy of the transition state (1–2 and 1–3)[22]. The mean rate for each shift quickly converges for an increasing number of barrier predictions per run (Fig. 3d, see the Supplementary Methods for further discussion). Thus, KIMMDY robustly reproduces the overall reaction dynamics, with only slight deviations for the most prominent shifts relative to each other.

The overall correct ranking and robustness of HAT reactions is also evident when directly comparing absolute reaction rates (Fig. S2),

and considering that the selection probabilities remain approximately stable across the 20 independent KIMMDY simulation runs in Fig. 3c. However, the GNN-based emulator underestimates the absolute rates. A reason for this could be the training data of the GNN-based emulator, which included constrained transition states for which only directly involved atoms were optimised[18]. Additionally, quantum mechanical tunnelling is ignored, which is known to increase rates by up to two to three magnitudes at 300 K[23]. Including zero-point vibrational energy corrections in the training data set of the emulator, as well as fine-tuning of the model to alkyl radicals, might further improve the results. Notwithstanding this underestimation, KIMMDY accurately reproduces the expected reaction probabilities, the decisive quantity in the application scenario of our reaction emulator. Also, as demonstrated by our validation, KIMMDY with learned barriers can be readily applied even to systems (*n*-alkanes) not explicitly included in the training data set (proteins).

## KIMMDY reveals reaction pathways for protein radicals
Having validated our reaction emulator for HAT, we move on to the first application in a biological system. Here, radical species are the product of homolytic cleavage and subsequently migrate through the system in several consecutive HAT reactions (Fig. 4a).

Collagen, formed from aligned and crosslinked triple helices (Fig. 4b), harbours radicals from homolytic cleavage when subjected to mechanical stress[24]. Migration of these mechanoradicals to DOPA, a post-translationally modified catechol-containing amino acid, was proposed based on thermodynamic arguments and electron paramagnetic resonance (EPR) data[24,25]. While homolysis sites[26] and the role of DOPA as a radical scavenger[25] have been extensively studied, KIMMDY offers the capability to scrutinise radical transfer pathways from homolysis sites to radical scavengers.

In a collagen fibril system of 2.6 million atoms, we observe a plethora of radical transfer pathways within 600 emulated HAT reactions.

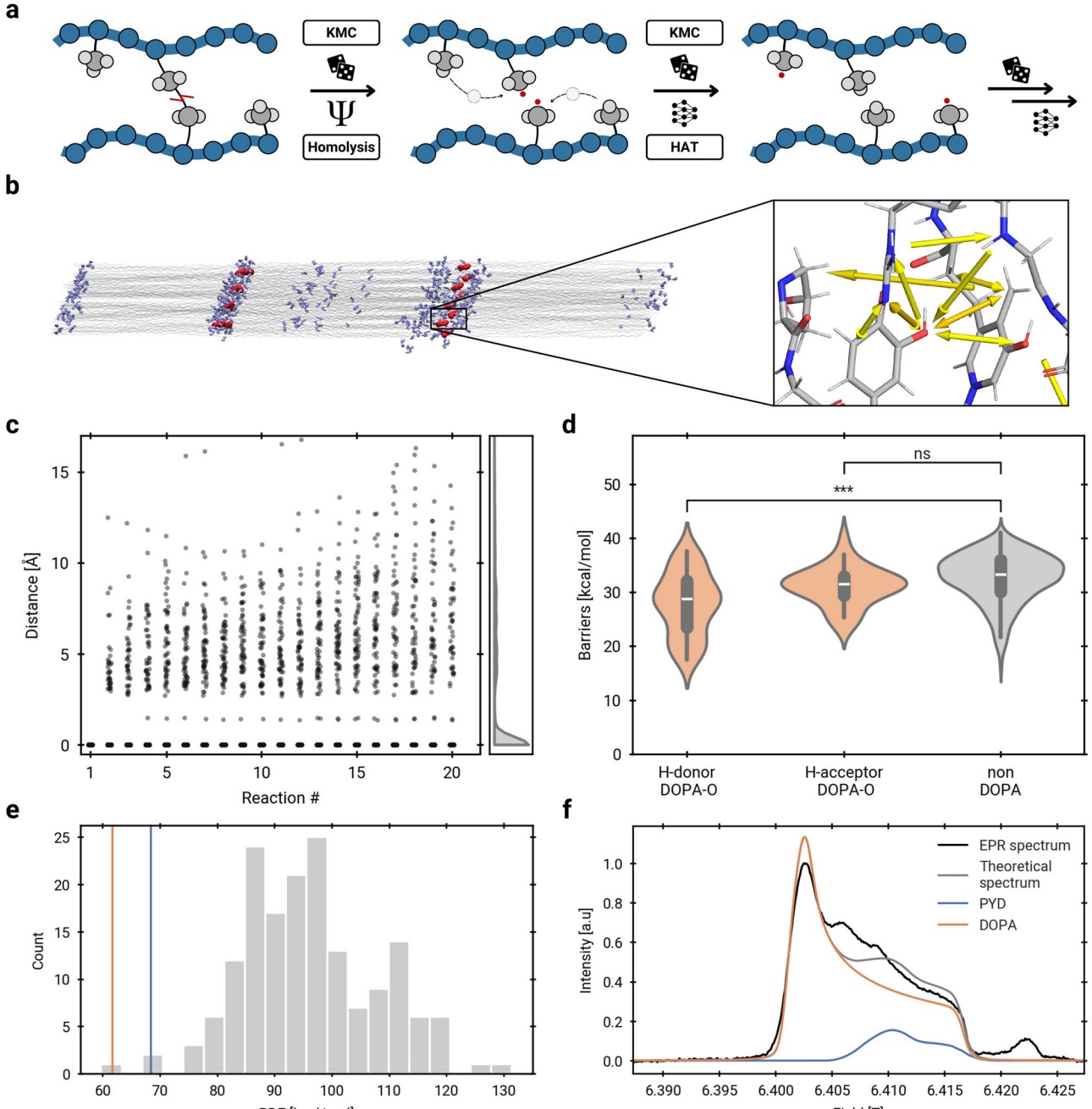

**Fig. 4 | DOPA and PYD scavenge homolysis-derived radicals. a** Consecutive reactions of radicals in KIMMDY are initiated with homolytic cleavage and followed by multiple HATs. Unpaired electrons are shown as red dots. **b** Structure of collagen fibril with backbone atoms (silver), potential DOPA sites (light blue) and PYD crosslinks (dark red). The zoom shows HATs as yellow arrows at a crosslink site. **c** Distance of radical atoms to the homolytically cleaved PYD C$\alpha$ and C$\beta$ atoms for 36 simulations with eight radical atoms each over a maximum of 20 consecutive HAT reactions. The density is shown before the 20th reaction. **d** Distribution of HAT barriers for reactions involving the DOPA hydroxy group as H-donor ($n = 49$, median=28.7 kcal/mol), H-acceptor ($n = 41$, median=31.4 kcal/mol) or no DOPA ($n = 467$, median = 33.2 kcal/mol). Boxes represent the interquartile range (IQR, Q1–Q3) with the centre line indicating the median and whiskers extending to the most extreme data points within 1.5×IQR from the quartiles. Significance of

differences was assessed using one-sided Welch's independent-samples $t$-tests without adjustment for multiple comparisons (*$p < 0.05$, **$p < 0.01$, ***$p < 0.001$). The differences between H-donor DOPA hydroxy group and non-DOPA are significant using Welch's $t$-test ($p = 3.9e-7$), whereas the difference between H-acceptor DOPA hydroxy group and non-DOPA are not significant ($p = 1.0$). **e** Bond dissociation energy BDE of PYD (blue line) and DOPA (orange line) in a distribution of BDE of abstractions processes in different amino acids (from ref. 27, grey bars). **f** Experimental EPR spectrum of rat tail collagen (black), reproduced with permission from ref. 25 (black) is compared with the spectrum of DOPA (orange), previously proposed as a key contributor to the signal. The spectrum of the PYD crosslink (blue), a candidate proposed by KIMMDY, can explain unresolved parts of the experimental data. Source data are provided as a Source Data file.

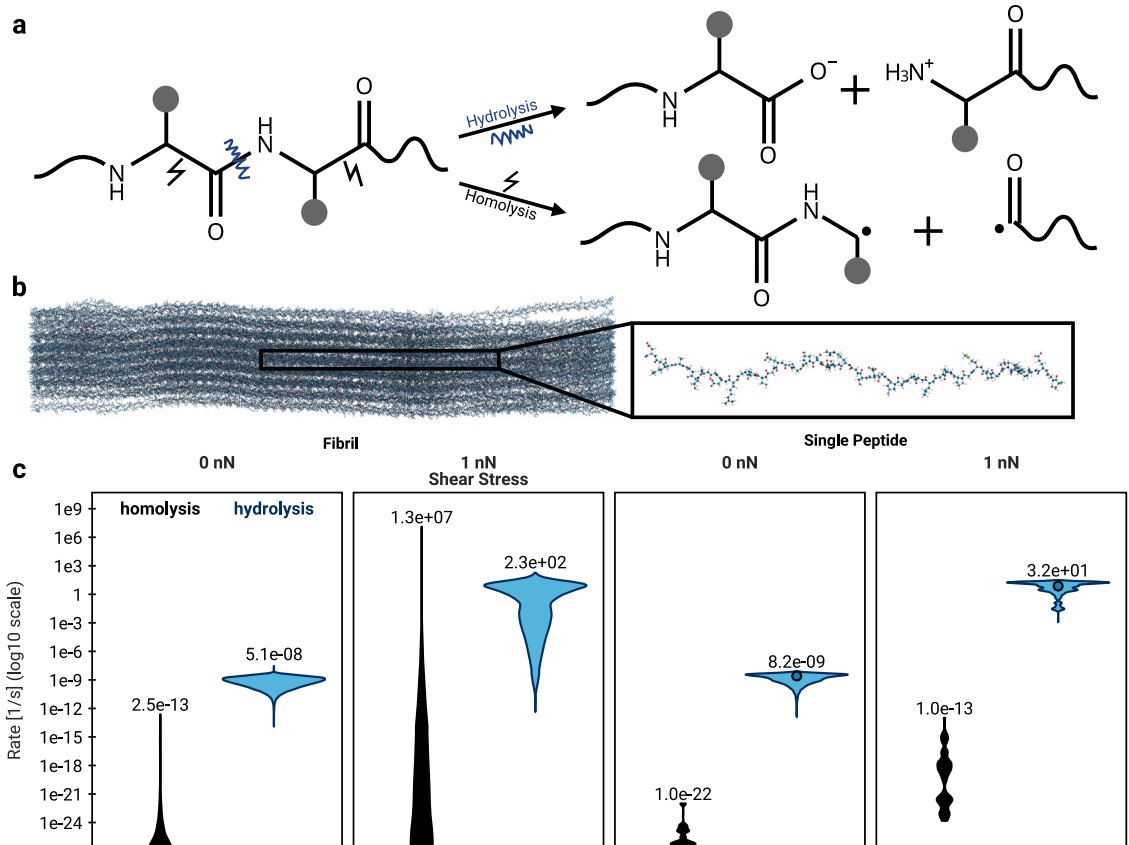

**Fig. 5 | Hydrolysis vs. Homolysis. a** Schematic representation of a peptide that undergoes either base-catalysed hydrolysis or homolytic cleavage. **b** Molecular render of a collagen fibril and a single peptide. **c** Comparison of the reaction rates for two competing reactions, homolysis and hydrolysis based on a physical model for homolysis[31] and a heuristic model[28] for hydrolysis. The value of the highest rate for each setup is shown above the respective distribution. The circles in the hydrolysis rate distributions of the single peptide denote the experimental reference value for the respective force. Source data are provided as a Source Data file.

Through HATs and conformational changes, radicals move tens of Angstroms away from the homolytically cleaved atoms (Fig. 4c). A wide range of amino acids host radical species during the simulations, and side-chain radicals were more frequent than backbone radicals (Fig. S11a, b). HATs are mainly intermolecular or with more than six bonds in between H-donors and H-acceptors and only include a few 1–2, 1–3, or 1–4 transfers, consistent with the above simulations of alkanes (Fig. S11c). As expected for a radical scavenger, barriers are significantly lower for reactions with a DOPA hydroxy group H-donor compared to those not involving DOPA (Fig. 4d). The median barrier difference is 4.5 kcal/mol, which, using the Eyring equation (Supplementary Notes), amounts to 1800 times faster reactions. We also sampled direct HATs from the homolytic cleavage site to the hydroxy group of DOPA (Fig. S11b). Thus, DOPA radical species are kinetically accessible in collagen under stress. We do not observe the DOPA hydroxy radical to be less reactive as a hydrogen acceptor compared to other protein radicals, and attribute this to rapid HATs between multiple adjacent radical scavengers (Fig. S11e, f).

Interestingly, a large population of radicals resulting from cross-link cleavage (C$\alpha$ and C$\beta$ atoms of the trivalent pyridinium-crosslink PYD) do not migrate during the reactive simulation (Fig. 4c). This indicates that they are relatively stable radicals, compared to more reactive H-acceptors formed in subsequent reactions which often migrate further. In a multiple hypothesis testing scenario, we tested for amino acid H-donors with a significantly lowered barrier (Fig. S11d). Here, the hydroxy group of the broken PYD is identified as a frequent H-donor. This moiety has not been investigated as a potential radical scavenger before, but is ideally positioned and kinetically accessible according to KIMMDY.

We asked if PYD can also thermodynamically serve as a radical sink in collagen by comparing its bond dissociation energy (BDE) for hydrogen abstraction to those in all 20 amino acids and DOPA[27] (Fig. 4e). Indeed, PYD and DOPA have the lowest BDEs, which implies that both are exceptionally good radical scavengers. A previously measured EPR absorption spectrum of stretched collagen[25] also supports the notion of PYD as a plausible radical candidate. It supports DOPA as the major radical species, but exhibits an otherwise unexplained region, which aligns with the computed PYD spectrum (see Supplementary Methods). Including PYD improves the overall fit to the experimental data, increasing the $R^2$ from 0.89 to 0.98 (Fig. 4f and Fig. S5). Thus, KIMMDY lets us reinterpret experimental data and identify an overlooked radical-stabilising moiety in collagen.

## KIMMDY can model competing reactions in biologically relevant systems

In most biochemical systems, different reactions can directly compete with each other, leading to vastly different products. To showcase the ability of KIMMDY to deal with different reaction types, we compare homolytic and heterolytic bond cleavage in collagen, but analogous simulations can be carried out for other biomolecular systems. While homolytic cleavage leads to subsequent chain reactions and detectable radicals (see section "KIMMDY reveals reaction pathways for protein radicals"), heterolytic cleavage is a closed-shell reaction and in case of proteins involves the attack of water, i.e. is a hydrolysis reaction (Fig. 5a). Both reactions can be promoted by force, both have been observed experimentally[24,26,28,29], and their competition depends on environmental effects, such as solvent accessibility or pH.

Direct comparisons between the two reactions using combined quantum mechanical and molecular mechanical simulations (QM/MM) are possible, but fundamentally limited by sampling and computational cost. This is where KIMMDY's modular system comes into play, providing an emulator for their competition in a realistic biological setting. The reaction rates for homolysis are modelled according to a physics-based Bell model[30,31] and those for hydrolysis via a heuristic model based on force-clamp experiments[28], pH, and surface accessibility, which demonstrates the ease of using KIMMDY even when conformation-dependent learned rates are not at hand.

As expected, in the absence of an external pulling force, both hydrolysis and homolysis yield low rates, that is, spontaneous fragmentation of peptides is unlikely at ambient conditions. When the protein systems are pulled at 1 nN, hydrolysis outcompetes homolysis for a single peptide chain. However, for a densely packed and cross-linked collagen system, with all else being equal, homolysis is accelerated drastically and reaches rates of similar orders of magnitude as hydrolysis and outcompeting it when comparing the highest rates. This is due to areas of high stress concentration in the large collagen fibril system that push the reaction rates into a region of the force response curve where hydrolysis already tapers off and homolysis still scales (See Fig. S13b and ref. [31]). This result is robust with regard to the model choice for hydrolysis (physical QM-based versus heuristic experiment-based) and other model parameters (see Supplementary Methods and Fig. S13).

Our results suggest that reactions that exhibit disparate scales of rates in simple molecules can become competitive in more complex molecular systems, with KIMMDY allowing direct conclusions on potential reaction outcomes, such as radical formation in the case of dense protein networks such as collagen.

### KIMMDY reveals unexpectedly low quantum yields in DNA origami motifs

KIMMDY can not only be applied to protein systems. Upon UV irradiation, pyrimidine bases in DNA can form cyclobutane pyrimidine dimers (CPDs), which play a crucial role in the development of skin cancer[32]. In DNA nanotechnology, CPDs are used as covalent crosslinks between staples, enhancing the chemical stability of DNA origami structures[33]. MD studies have suggested that the frequency of DNA conformations favourable for dimerisation can be directly related to the quantum yields $\phi$ observed in experiments[34–36], with KIMMDY now offering a direct test of this scenario.

A distance and angle-based heuristic model, with parameters tuned to reproduce experimental quantum yields for small benchmark systems, was used to calculate reaction rates through KIMMDY (see Supplementary Methods). Simulation frames with rates above a defined threshold are counted as dimerisable configurations contributing to the quantum yield.

In the thymine dinucleotide (TdT) system, KIMMDY predicts two products, the expected *cis-syn* but also the *trans-syn* CPD isomer, albeit at much lower quantum yields (Fig. 6a and Fig. S7). Inspection of the underlying conformation shows that prior to the reaction, the thymine bases can undergo a *syn-anti* transition by rotation around the N-glycosidic bond (Fig. S7a, b). In experiments, the *cis-syn* isomer forms eight times more frequently than the *trans-syn* isomer during TdT irradiation[37]. Our KIMMDY results put forward a lower quantum yield, together with the *syn-anti* precursor state being less frequently populated, as an explanation.

KIMMDY predicts quantum yields for overhang and crossover structures (Fig. 6b) that are substantially lower than for double-stranded (ds) or nicked motifs (Fig. 6d), despite overhangs and crossovers being commonly used to stabilise DNA origami via CPD crosslinking. This discrepancy may be explained by three reasons: long irradiation times used for origami stabilisation could compensate for the inherently low quantum yields; the distance and dihedral angle

distributions show (Fig. 6c) an increased flexibility of the crossover system, which might be more restricted within the full origami structure; other photoproducts such as *cis-anti*, *trans-anti*, and 6-4 photoproducts (see Supplementary Methods), which are not accounted for in the current KIMMDY plugin, might be formed. We therefore propose that future experimental studies should investigate both the quantum yields and the diversity of photoproducts in DNA origami.

In systems with consecutive dimerisation sites (Fig. 6b), we observe no change in predicted quantum yields (Fig. 6e). This indicates that a close-by CPD separated by five base pairs from a second reactive site does not significantly affect the conformational space available for an additional dimerisation. This insight suggests that DNA previously damaged by CPD formation is not necessarily more susceptible to additional dimerisation, which may have implications for the formation of mutagenic lesions.

## Discussion

KIMMDY offers the emulation of single reactions, reaction cascades, and competing reactions throughout molecular systems at a computational effort which is many orders of magnitudes smaller than alternative approaches, being it at the ab initio, hybrid QM/MM, reactive force field, or MLIP levels. Direct simulations with the latter and variations thereof are prohibitively slow for most biochemically relevant processes, as they are currently limited to the microsecond (or even slower) timescale and require enhanced sampling for selected reactions. KIMMDY instead emulates reactions, i.e. does not explicitly model them, thereby reaching arbitrary timescales at low computational cost.

The novelty of our work lies in harnessing the power of recently developed graph neural networks for two new tasks: (i) for emulating reactions and their barriers in a conformation-aware manner using a hybrid MD/ML approach and (ii) for learning a molecular mechanics force field as required for the physics-based ensemble generation in a reactive setting. We validated our reaction emulator by comparison to experimental and DFT data.

We recognise parallels to methods leveraging Markov state models (MSMs) to describe reaction kinetics[38]. In contrast to these approaches, KIMMDY does not require computationally expensive ab initio or MLIP MD simulations, but instead allows the use of MM force fields. This substantially reduces the computational cost and enables the investigation of larger biomolecular systems, such as the collagen fibrils presented here. Additionally, KIMMDY is readily applicable to systems with similar chemistry without the need for prior computations. However, KIMMDY limits the accessible reaction space to reactions predefined by the reaction plugins, whereas ab initio methods can discover all possible reactions and transition states.

In our three exemplary applications, KIMMDY yields unexpected predictions to reinterpret former and inspire new experiments. First, we identify a new efficient radical-stabilising moiety in collagen, which can explain hitherto unexplained spectroscopic data and is relevant for our understanding of tissue mechanochemistry and ageing. Secondly, KIMMDY shows that the competition between open and closed shell biochemistry can substantially shift when going from simple to more complex structural environments, reconciling apparent contradictions of previous observations. Thirdly, KIMMDY predicts quantum yields of photoinduced dimerisation in DNA to be unexpectedly low in common DNA origami motifs and to be independent of previous nearby dimerisations—testable findings that are highly relevant to biomedical applications.

Although KIMMDY is able to simulate a diverse set of systems and reactions, there are inherent limitations. KIMMDY reactions are selected and implemented for specific reaction types, it cannot predict reaction types not anticipated by the researcher. For example, dimerisation could be extended to account for all possible photoproducts. Exploring reaction networks more widely[39] prior to

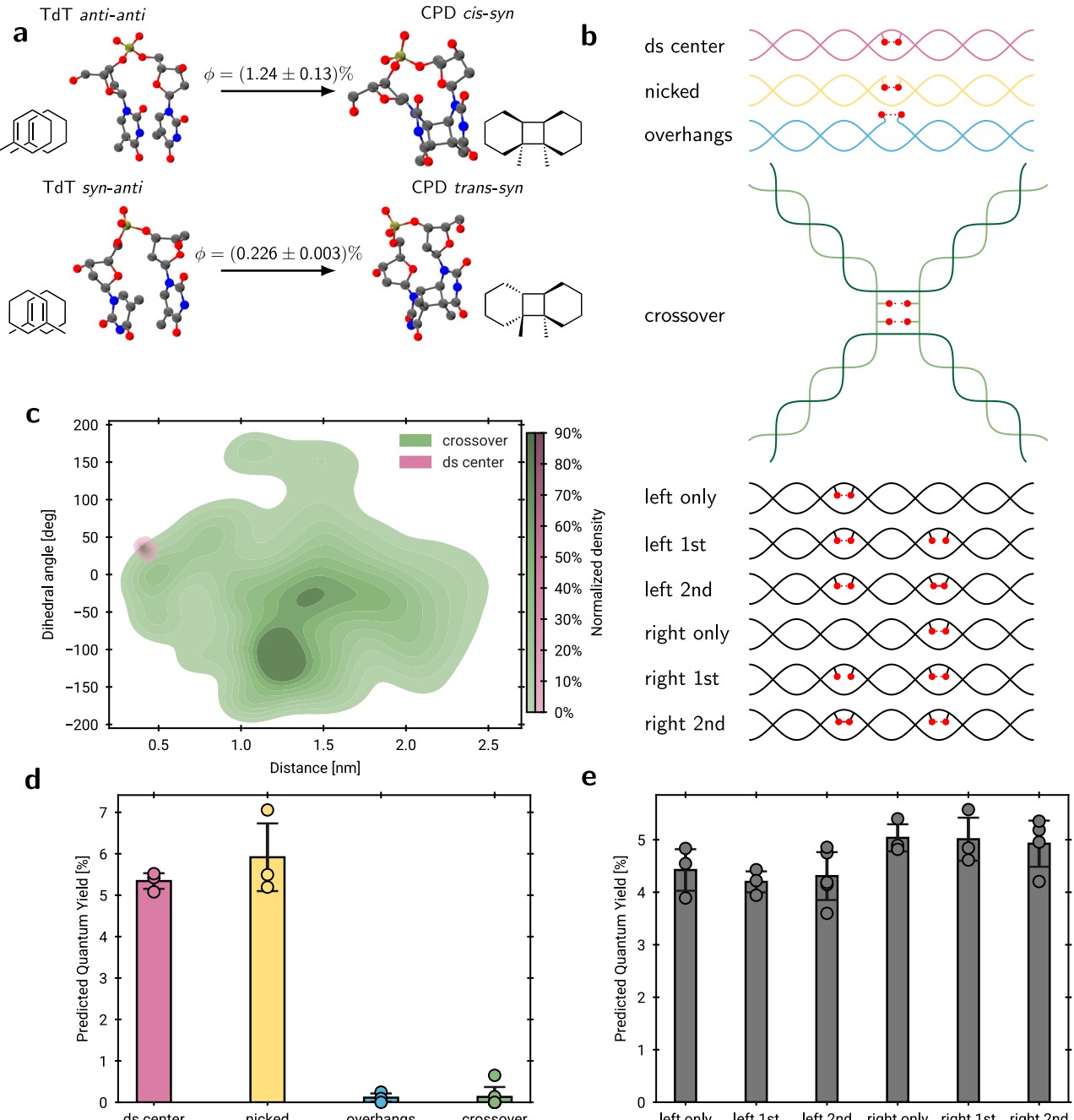

**Fig. 6 | DNA systems investigated with KIMMDY. a** Reaction scheme of thymine-thymine dimerisation starting from the *anti-anti*-conformation or the *syn-anti*-conformation, yielding *cis-syn* and *trans-syn* diastereomers with different predicted quantum yield $\phi$. **b** Structural schemes of tested DNA motifs. Red circles denote thymines. Solid connections between thymines indicate a dimer; dashed connections between thymines denote a reactive site considered by KIMMDY. **c** Kernel density estimate (KDE) of distance and angle distributions of ds centre and crossover systems determined from 10,000 snapshots randomly sampled from three MD simulations (see Supplementary Methods), respectively. The KDE is plotted over the distance between reactive thymine double bonds and the dihedral angle between them. **d** Predicted quantum yields for motifs commonly found in DNA origami ($n = 3$ for ds centre, nicked, and overhangs, and $n = 6$ for crossover from separate 100 ns MD simulations). Bars represent mean ± s.d.; dots indicate individual simulations. **e** Predicted quantum yields for consecutive dimerisation systems (and respective control systems), in which the two reactive sites are five base pairs apart ($n = 3$ for left only, left first, right only, right first, $n = 4$ for right second, and $n = 5$ for left second from separate 100 ns MD simulations). Bars represent mean ± s.d.; dots indicate individual simulations. Source data are provided as a Source Data file.

KIMMDY, can lift this limitation in the future. Prospective users must obviously be aware of the relevant limitations of the underlying ML model with regard to uncertainty and transferability. Finally, as shown in the *n*-alkyl case, HAT rates did not take tunnelling into account. Even though the probability of choosing a reaction is accurate, the time to the next reaction is off by a factor, limiting the interpretation of

reaction times and their use when it comes to competing reactions. KIMMDY, however, offers the incorporation of such as tunnelling or other (quantum) effects into the emulator, which is subject to future work.

Heuristic models make the implementation of new reaction types straightforward, but their predictive power is limited. An example is

the simulation of reactions in coarse-grained models, where the state of the art still largely relies on heuristic approaches[40–42]. Emulating biomolecular reactivity in coarse-grained simulations such as those using MARTINI while approaching DFT accuracy can be achieved with KIMMDY by using a machine-learning engine that transfers reaction rates from all-atom to coarse-grained representations, which will be the subject of future research. As another example for substituting heuristic rates by learning from ab initio data, dimerisation rates in DNA can be estimated on the basis of transition states on the excited-state energy surface, which should increase the accuracy of the emulator[14]. The hydrolysis model uses a linear relation to the solvent-accessible surface area, again an assumption that an ML-based emulator can lift. Still, KIMMDY is applicable to problems for which efficient and sufficiently accurate heuristic models can be identified.

To conclude, KIMMDY is a versatile and highly extensible computational method to explore biochemical reactivity amidst the fluctuations of a dynamic molecular system. Large-scale simulations with KIMMDY include biochemical reactivity at little extra cost. We foresee KIMMDY to deliver new insights at many frontiers, from how enzymes work under tight regulation within the cell, to how biomolecules are permanently modified, damaged, degraded or repaired.

## Methods

### Adaptive kinetic Monte Carlo

KIMMDY simulates the time evolution[43] of reactive biopolymer species. It uses the rejection-free kMC (rf-kMC) algorithm[44,45], which consists of the following steps: (1) From an initial state, create an event list of all possible transitions, (2) Draw a uniform random number $u_1$ and select the event for which $F(p_{i-1}) < u_1 \leq F(p_i)$, where $F$ is the cumulative function and $p_i$ the probability of event $i$. (3) Draw a uniform random number $u_2$ to update the time according to $\Delta t = -\frac{\ln u_2}{R}$, where $R$ is the sum of all rates. (4) Carry out the event $i$. In KIMMDY, the simulated events are chemical reactions, and the transition rates are reaction rates.

Adaptive kMC[15] is a variation of rf-kMC where the event list for a state is calculated only if it is populated during the kMC simulation. This approach is beneficial if the state space is too vast or the number of possible events per state is too large to precompute the event probabilities. In biopolymers and in general soft matter systems, most atoms exhibit reactivities because they are embedded in a certain structural context with electrostatics, solvent accessibility and steric effects. This dependence on the environment necessitates calculating reaction rates individually for every set of reactant atoms, rendering their reactive simulation an ideal application for adaptive kMC.

In KIMMDY, a kMC step is divided into three tasks (Fig. 2a, see Supplementary Methods). First, possible reactions are sampled by generating a conformational ensemble of the current state, which then is used to predict reaction rates to neighbouring states either by a heuristic, physical or machine learning model (Fig. 2b). The second task comprises selecting the event and a corresponding update time according to the kMC algorithm. Finally, MD simulation parameters and coordinates are adapted to obtain the product state. Several kMC steps can be chained in a cyclic process. We built KIMMDY to be modular, flexible and extensible.

### Sample reaction

For reaction sampling, a conformational ensemble serves as the input to the reaction emulator, which predicts reaction rates for transitions to chemical states different from the current ensemble. In the applications presented here, a molecular system described by coordinates and simulation parameters is simulated using Molecular (MD) or Stochastic Dynamics (SD), but any other ensemble generator can be used, e.g. a learned one[46].

One approach to calculate the event list from the conformational ensemble generated with MD or by other means is by using ensemble averages of properties and relating them to reaction rates by physics-based or empirical models. Alternatively, we use a machine-learned model to predict transition rates from individual snapshots to calculate a constant average rate per reaction over the whole ensemble (see Supplementary Notes). This has the additional benefit of accounting for entropic effects by sampling how frequently highly reactive conformations are visited.

For the HAT application, using the conformation ensemble instead of a single structure representing a state, the prediction task is significantly simplified from emulating a complete transition state search to a local optimisation of the reacting atoms. Still, the MD sampling problem may lead to an underestimation of reaction rates.

In this study, the sampling time and number of trajectory frames are adjusted for the respective modelled reactions depending on the convergence of rates (see Supplementary Methods). The simulation setup is automated within KIMMDY and relies on user-supplied simulation parameters. Simulation details for the molecular systems used in this study are described in the Supplementary Methods. KIMMDY is designed as a framework to be extended to diverse reactions. To this end, a plugin architecture providing a stable interface is available.

### Choose reaction

From the collected event list, a reaction with a probability directly proportional to its rate is chosen. KIMMDY then associates with this event a time update from all predicted reactions. We implemented different kMC algorithms in a modular fashion in KIMMDY. In this work, only the rf-kMC algorithm with adaptive event list generation is used.

### Effect reaction

To effect the chosen reaction, the corresponding reaction recipe is applied to change the molecular system topology, parameters and coordinates to the product state. Recipes define reactions through elementary recipe steps. Bind and break reference the two involved atoms and modify the MD bond definitions. Angles, dihedrals and pairs are modified accordingly. Changes in force field parameters are either handled by supplying a force field that has parameters for all reaction products or by re-parametrising the bonded parameters with the generally applicable machine-learned Grappa force field[21] combined with heuristics for the nonbonded parameters (see Supplementary Methods). To generate the product coordinates, Place moves an atom in a certain snapshot to a new position and, as an alternative, Relax starts a MD simulation with the slow-growth feature of GROMACS to interpolate smoothly between reactant and product parameters. Finally, an equilibration MD simulation is performed to relax the new topology and reach an equilibrium ensemble as input for the next reaction sampling step. The equilibration timescale should be sufficient to ensure sampling for the next reaction from the product state Boltzmann distribution. Thus, KIMMDY models state transitions as a Markov process.

### Efficiency of KIMMDY

With KIMMDY, simulations of consecutive and competing reactions in large biopolymers are possible for the first time. Taking the collagen fibril system with 2.6 million atoms as an example, tens of thousands of reaction rates are predicted per kMC step using physics-based or empirical models. For the machine-learned HAT model, about 100 reactions are evaluated every kMC step. While the necessary sampling and evaluation time depends on the rate prediction model, we observe that most time is spent on MD sampling for the aforementioned applications, and that the overhead introduced from the KIMMDY framework is negligible. Overall, one kMC step for the collagen fibril HAT application takes 10 h on a consumer GPU (Fig. S12a). Comparing this duration with calculating a single reaction barrier by DFT, as done for the HAT GNN training, our emulator is roughly 100 times faster and

at the same time accounts for conformational changes of the molecular system. Other systems see even greater speed-ups. For example, evaluating the reaction barrier of a reaction as complex as base-catalysed hydrolysis with DFT in a QM/MM scheme requires extensive sampling and would take upwards of one year on a single CPU for a single reaction site. But with the hydrolysis reaction rates based on a heuristic, querying for all possible reactions is done in seconds.

## Reporting summary

Further information on research design is available in the Nature Portfolio Reporting Summary linked to this article.

## Data availability

The datasets generated and/or analysed during the current study are available via Zenodo (https://doi.org/10.5281/zenodo.18765302). This repository contains simulation input files, initial and final molecular dynamics configurations, custom force field parameters, analysis scripts, and processed data necessary to reproduce all reported results. Due to their size (several terabytes), full molecular dynamics trajectories are not deposited in the public repository. Full trajectories are available from the corresponding authors upon request. The Rattus norvegicus type I collagen fibril structure (PDB: 3HR2, https://doi.org/10.2210/pdb3hr2/pdb) was used in this study. Source data are provided with this paper.

## Code availability

KIMMDY is publicly available at https://github.com/graeter-group/kimmdy. The various KIMMDY plug-ins, e.g. for homolysis, hydrolysis, HAT and DNA crosslinking reactions, can be found under the kimmdy tag: https://github.com/topics/kimmdy. Data collection was performed using GROMACS v2023.3[19], PLUMED v2.9.0[47,48], KIMMDY v8.0.0, GRAPPA v1.4.1-radical[21], ORCA v5.0[49,50] and EasySpin v6.0.6[51], while data analysis was carried out with Python v3.10.18 and KIMMDY v8.0.0.

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

## Acknowledgements

This work was supported by the Klaus Tschira Foundation (to F.G. and K.R.). This project has received funding from the European Research Council (ERC) under the European Union's Horizon 2020 research and innovation programme (grant agreement No. 101002812) (to E.H., J.B. and F.G.). D.S and F.G. gratefully acknowledge the support of the Klaus Tschira Foundation (SIMPLAIX project 8). D.S. and F.G. gratefully acknowledge the provision of computing resources by the state of Baden–Württemberg through bwHPC and the German Research Foundation (DFG) through Grant Nos. INST 35/1597-1 (Helix cluster). We acknowledge funding through the Deutsche Forschungsgemeinschaft (DFG, German Research Foundation) under Germany's Excellence Strategy–2082/1–390761711. The authors gratefully acknowledge support by the German Research Foundation (DFG) grant GRK 2450 (to B.N.S. and F.G.). This research was conducted within the Max Planck School Matter to Life, supported by the German Federal Ministry of Education and Research (BMBF) in collaboration with the Max Planck Society (to B.N.S. and F.G.). The authors would like to thank Alexander I. Jordan and Alice Allen for helpful discussions and Alexander I. Jordan for suggestions regarding the statistical background of the methodology.

## Author contributions

E.H.: Conceptualisation, data curation, formal analysis, investigation, methodology, software, project administration, validation, visualisation and writing (original draft, review and editing). J.B.: Conceptualisation, data curation, formal analysis, investigation, methodology, software, project administration, validation, visualisation and writing (original draft, review and editing). K.R.: Conceptualisation, data curation, formal analysis, methodology, software and writing (review). E.U.: Formal analysis, investigation, methodology, visualisation and writing (original draft, review and editing). B.N.S.: Formal analysis, investigation, methodology, software, visualisation and writing (original draft, review and editing). D.K.: Formal analysis, investigation, software, visualisation and writing (review & editing). D.S.: Data curation, formal analysis, investigation, methodology, visualisation and writing (original draft, review & editing). C.A.-S.: Conceptualisation, formal analysis, methodology and writing (review & editing) F.G.: Conceptualisation, methodology, project administration, resources, supervision and writing (original draft, review & editing).

## Funding

## Competing interests

The authors declare no competing interests.
