## [Transparent Peer Review file · Nature Communications]

KIMMDY: a biomolecular reaction emulator

Corresponding Author: Professor Frauke Gräter

Version 0:

Reviewer comments:

Reviewer #1

(Remarks to the Author)

The current platform has for sure merit and combines different methods. It tries to show a broad applicability. I do have still different comments.

055 check grammar ", Nature"

can not should be cannot

Part before 090: please add a figure on typical interactions (and then captured by models and what here is, so bridge to Figure 1)

MD and kMC combinations, one has MD to kMC and reverse, e.g. Nat. Mater. 2021 20, 1422. Perhaps good to make a general reader aware.

HAT 500 K. The aspect of T or in general Arrhenius needs to be covered, so how "heuristic" or "fundamental" is this approach. Specifically also how reaction rates are obtained.

Figure 2 is interesting (and there is SI/Methods) but a general reader will not capture how the approach works. This needs be explained or the work will be seen as too much generating results. A more detailed flow chart (discussion) is needed in the main text. Also the benchmark systems need more framing for a general reader.

A very positive aspect is that many applications are shown. A bit of better framing would help to increase readability of the selected journal.

How relevant is the tunneling in a general context of reactions.

Another general comment: can more comparison be included with "competitive" software. So is there a reactive system to be included.

(Remarks on code availability)

not applicable

Reviewer #3

(Remarks to the Author)

The authors describe a program package, called KIMMDY, which can perform simulations for large bio-systems to produce rate constants for various reactive processes. I think the results are significant and will capture the interest of the specialist in the field. This study may deserve publication in Nature Communications.

Is KIMMDY an acronym? The meaning of the name should be clarified.

In the SI there are several typos and formatting issues:

"an internal Topology object is build"

"A checkpoint system facilitates long KIMMDY simulations (Fig. ??)."

At Tables S1-S4 captions should be above the tables.

"However, those percentages change in independent measurements of different tissues (as shown in the Fig. ??)"

"dynamics for 100 ns"

"rate of 0 1/s"

"ions were Na^+ and Cl^- "

"h-bonds"

"gaussian distribution" should be "Gaussian distribution"

"morse potential" should be "Morse potential"

"Ising spin model [?]"

"Henkelman and Jónsson [?]"

"Similarly, we refer the reader to [?]"

"inverse transform sampling (cite)"

(Remarks on code availability)

Reviewer #4

(Remarks to the Author)

The authors present KIMMDY, a multi-scale framework that addresses the challenge of simulating biochemical reactions by integrating Molecular dynamics (MD) simulations, kinetic Monte Carlo (kMC), and graph neural networks (GNNs), thereby bridging the gap between the high resolution (atomistic) needed to describe the reactants and the relatively long timescales (reaching seconds) of the reactions, for realistic systems. Specifically, a short MD simulation with a given force field is used to sample an ensemble of conformations, possible chemical reactions are identified, a fast emulator is used to predict the reaction rates using GNNs, and kinetic MC is used to select reactions stochastically. The molecular system is modified by executing the chosen reaction. The application of this method to complex events, such as radical migrations and photochemical dimerization, and its comparison with experiments demonstrate its potential to bypass the prohibitive costs of quantum-level simulations.

The proposed framework successfully addresses a critical bottleneck in computational chemistry and computational biology. Conventional fixed-topology molecular dynamics cannot model chemical reactions by design, and even if it could, it would be limited by computational inefficiency. Hybridizing MD for conformational sampling with kMC for rare-event sampling is a neat strategy to address this limitation. The authors implemented this approach, tested it across several systems, and directly compared it with experimental results. These results are overall impressive, though we have several suggestions for improving this work, in particular regarding comparisons and/or contextualization of alternative approaches, and calibration of predictive uncertainty and bias.

Specific comments

The authors argue that although ML interaction potentials such as ANI, SchNet, MACE, and NequIP are reactive by design, they suffer from the same fundamental problem as MD simulations: they are too slow to model reactions. However, while these methods do not address the challenge of capturing rate events, they are reported to provide faster per-step running times compared to classic MD. Some have been used to simulate reactions, such as the chemical reaction of Lithium Phosphate in the NequIP paper. If feasible, the authors should directly compare KIMMDY performance, e.g., using NequIP for the same reaction. If a direct comparison is prohibitively time-consuming, the authors should at least discuss this point critically.

Simulations can also be accelerated by coarse-graining, which both reduces the system's dimensionality and smooths the energy landscape, thereby extending the simulation time. For example, reactive MARTINI (Sami and Marrink, JCTC 2023) has been used to model reactions using a coarse-grained model. Is KIMMDY faster with a traditional force field than with a reactive coarse-grained potential, such as reactive MARTINI? If technically feasible with a reasonable effort, the authors should directly compare the two, at least on a limited scale. If it is a considerable feat, it is reasonable to settle for contextualizing coarse-grained reactive potentials in the Discussion. Of course, for coarse-graining potentials, this is also a matter of decreased resolution (but that does not necessarily matter if the objective is inference of reaction features).

In principle, KIMMDY can also be used in combination with coarse-grained or ML potentials, dramatically speeding up the MD ensemble generation step. The authors briefly mention the “near-future plugging in” of ML potentials in Supp, but we consider this a point worth raising in the Discussion, for both coarse-grained and ML potentials (while considering the differences in resolution).

Markov State Models have been recently used to describe reaction dynamics (<https://www.pnas.org/doi/10.1073/pnas.2406356121>). At least conceptually, there is some analogy between that work and the proposed methodology (without taking away from its novelty - these works are quite different), as MSMs are used to describe the high-level kinetics. In contrast, short simulation trajectories are used to describe the fine details. The authors may contextualize such alternative approaches in the Discussion.

Optional suggestion: The manuscript acknowledges that the GNN tends to underestimate absolute rates (attributed partly to missing nuclear quantum effects like tunneling). While the authors have performed a Brier divergence analysis, they could consider validating the barrier's calibration more rigorously. Specifically, if the model predicts a high probability of a reaction pathway, does this align with ground-truth frequency? A calibration plot comparing predicted vs. observed barrier distributions would be valuable. For example, in the n-alkyl radical benchmark (Fig. 2c, Fig. A1), the authors could compare the model-predicted barrier (or rate) distributions against the experimentally inferred distributions. For each HAT shift type, does the predicted selection probability match the experimental one within confidence intervals?

Please perform at least a limited sensitivity analysis of the barrier predictions with respect to input noise. Do minor fluctuations in the input geometry (thermal noise) cause significant variances in the predicted barrier ranking? Discuss how robustness (or sensitivity) of the GNN to conformational noise may influence the kMC steps, to ensure these steps are physically meaningful rather than artifacts of model instability.

While the cross-domain generalization of GNNs is briefly demonstrated (the n-alkyl radical HAT benchmark), the manuscript could benefit from a practical guide and an example for users applying KIMMDY to systems that differ significantly from the training data. As a potential user, one needs to understand the operational limits and whether and how KIMMDY can be applied 'out of the box' to new chemistries, or whether strictly localized retraining is always required. We leave details to the discretion of the authors, but some examples are as follows:

- How can a user verify if the pre-trained GNN is valid for their specific, novel biological system?
- The authors could explicitly quantify how performance degrades as the chemical environment deviates from that of the training set.
- Does the model provide a confidence score or an uncertainty metric to warn the user when the reaction environment is too distinct to model accurately?

(Remarks on code availability)

We have only briefly reviewed the code, as we were allocated a short timeframe for the review. However, the code appears to be well-maintained and in good shape. We did not have the time to test it ourselves, though the installation seems quite straightforward.

Reviewer #5

(Remarks to the Author)

(Remarks on code availability)

Version 1:

Reviewer comments:

Reviewer #1

(Remarks to the Author)

I have carefully checked the revised manuscript and the reply letter. The work has been sufficiently improved in view of publication.

(Remarks on code availability)

Reviewer #3

(Remarks to the Author)

I think the present version of the manuscript can be accepted.

Minor comments on the SI file:
The title of Table S4 should also go above the table.
At Fig. S11 caption "Tableau" should be "Table".

(Remarks on code availability)

Reviewer #4

(Remarks to the Author)
ok with current version

(Remarks on code availability)
commented in previous review

Reviewer #5

(Remarks to the Author)
I co-reviewed this manuscript with one of the reviewers who provided the listed reports. This is part of the Nature Communications initiative to facilitate training in peer review and to provide appropriate recognition for Early Career Researchers who co-review manuscripts.

(Remarks on code availability)
I only glanced at the code
I did not assess the code

Response to Referees

Please find the below point-by-point answers. Original reviewer comments indented and *italic*.

Reviewer 1

The current platform has for sure merit and combines different methods. It tries to show a broad applicability. I do have still different comments.

We thank the reviewer for acknowledging KIMMDY's merit and applicability,

055 check grammar “, Nature” can not should be cannot

We have corrected those.

Part before 090: please add a figure on typical interactions (and then captured by models and what here is, so bridge to Figure 1)

This was a very helpful suggestion. We have introduced a new Figure 1 which highlights the general idea of KIMMDY, juxtaposes it with QM/MM and in this way also schematically shows the type of interactions captured for the kMC step.

MD and kMC combinations, one has MD to kMC and reverse, e.g. Nat. Mater. 2021 20, 1422. Perhaps good to make a general reader aware.

We thank the reviewer for pointing us to this related work, which we added to the introduction, together with Gilley et al., chemrxiv, Oct 2025 (Gilley, Sathyaseelan, and Savoie 2025), which has been published while this manuscript was under review. De Keer et al. run MD after a network topology has been built by kMC (De Keer et al. 2021). Gilley et al. use ‘intrinsic’ rates which are scaled by the diffusion needed in the MD-simulated systems. Both methods are quite distinct from KIMMDY, which considers reactant relative conformations, as proposed by MD, for the rate prediction.

HAT 500 K. The aspect of T or in general Arrhenius needs to be covered, so how “heuristic” or “fundamental” is this approach. Specifically also how reaction rates are obtained.

The reaction rates are obtained using the Eyring equation motivated by transition state theory (TST), which is theoretically grounded. The barriers used for the Eyring equation however, are predicted using a machine learning model published with inherent prediction errors as is typical for any regression model (see reference Riedmiller et al. (2024), for further details of the model architecture used) and can therefore be considered “heuristic”. Lastly, the MD is run using a classical force field which will additionally introduce heuristic errors.

The temperature T enters through the MD settings as well as through the Eyring equation (SI eq. 26 and 27).

In the paper we mainly refer to “heuristic” as meaning “experiment-derived”. Notwithstanding, some heuristic, in the wider sense, always go into our model. We now clarified this in caption of fig. 2.

Figure 2 is interesting (and there is SI/Methods) but a general reader will not capture how the approach works. This needs to be explained or the work will be seen as too much generating results. A more detailed flow chart (discussion) is needed in the main text. Also the benchmark systems need more framing for a general reader.

Thank you for making us aware and for the helpful suggestion. We have added more details and explanations to (now) fig. 3.

A very positive aspect is that many applications are shown. A bit of better framing would help to increase readability of the selected journal.

We thank you for recognising our effort to include many applications to show the flexibility of KIMMDY. The revised manuscript attempts to tie them together with more purpose.

How relevant is the tunneling in a general context of reactions.

Thank you for raising this point. Tunneling is especially relevant in the case of Hydrogen Atom Transfer (HAT), due to its light mass [Chem. Soc. Rev., 2017,46, 7548-7596, doi.org/10.1039/C7CS00602K]. We added a plot to the SI (sec. 2.3) to illustrate that tunnelling is especially noticeable at lower temperatures and decreases as temperature increases, but overall is similar for the 1-4, 1-5 and 1-6 case.

Another general comment: can more comparison be included with “competitive” software. So is there a reactive system to be included.

We expanded on this, especially in the discussion. In short, the main point to stress here is one of timescales. KIMMDY can simulate reactions with timescales far outside the reach of other reactive methods (even microseconds to seconds) due to the kinetic Monte Carlo approach, while at the same time profiting from dynamic sampling with fast classical MD simulations.

Reviewer 3

The authors describe a program package, called KIMMDY, which can perform simulations for large bio-systems to produce rate constants for various reactive processes. I think the results are significant and will capture the interest of the specialist in the field. This study may deserve publication in Nature Communications.

We are grateful to Reviewer 3 for considering our contribution ‘significant’ and interesting to the field.

Is KIMMDY an acronym? The meaning of the name should be clarified.

We have clarified what KIMMDY stands for (KInetic Monte carlo Molecular DYnamics)

In the SI there are several typos and formatting issues: “an internal Topology object is build” “A checkpoint system facilitates long KIMMDY simulations (Fig. ??).” At Tables S1-S4 captions should be above the tables. “However, those percentages change in independent measurements of different tissues (as shown in the Fig. ??)” “dynamicsfor 100 ns” “rate of 0 1/s” “ions were NA+ and Cl-” “h-bonds” “gaussian distribution” should be “Gaussian distribution” “morse potential” should be “Morse potential” “Ising spin model [?]” “Henkelman and Jónsson [?]” “Similarly, we refer the reader to [?]” “inverse transform sampling (cite)”

We have corrected these issues, thank you for spotting them.

Reviewer 4

The authors present KIMMDY, a multi-scale framework that addresses the challenge of simulating biochemical reactions by integrating Molecular dynamics (MD) simulations, kinetic Monte Carlo (kMC), and graph neural networks (GNNs), thereby bridging the gap between the high resolution (atomistic) needed to describe the reactants and the relatively long timescales (reaching seconds) of the reactions, for realistic systems. Specifically, a short MD simulation with a given force field is used to sample an ensemble of conformations, possible chemical reactions are identified, a fast emulator is used to predict the reaction rates using GNNs, and kinetic MC is used to select reactions stochastically. The molecular system is modified by

executing the chosen reaction. The application of this method to complex events, such as radical migrations and photochemical dimerization, and its comparison with experiments demonstrate its potential to bypass the prohibitive costs of quantum-level simulations.

We thank you for this insightful and positive evaluation of our contribution.

The proposed framework successfully addresses a critical bottleneck in computational chemistry and computational biology. Conventional fixed-topology molecular dynamics cannot model chemical reactions by design, and even if it could, it would be limited by computational inefficiency. Hybridizing MD for conformational sampling with kMC for rare-event sampling is a neat strategy to address this limitation. The authors implemented this approach, tested it across several systems, and directly compared it with experimental results. These results are overall impressive, though we have several suggestions for improving this work, in particular regarding comparisons and/or contextualization of alternative approaches, and calibration of predictive uncertainty and bias.

We are grateful for your appreciation of our work as a ‘neat strategy’ and ‘overall impressive’. We address the concerns on the wider context and the calibration in detail below.

The authors argue that although ML interaction potentials such as ANI, SchNet, MACE, and NequIP are reactive by design, they suffer from the same fundamental problem as MD simulations: they are too slow to model reactions. However, while these methods do not address the challenge of capturing rate events, they are reported to provide faster per-step running times compared to classic MD. Some have been used to simulate reactions, such as the chemical reaction of Lithium Phosphate in the NequIP paper. If feasible, the authors should directly compare KIMMDY performance, e.g., using NequIP for the same reaction. If a direct comparison is prohibitively time-consuming, the authors should at least discuss this point critically.

We would like to clarify that MD simulations run with ML interatomic potentials (MLIPs) are slower than ‘classic MD’, that is, MD run with molecular mechanics force fields as most commonly done for biomolecules still to date. Our aim here is to go even beyond the timescale reachable by classic MD, while also being reactive. With kMC, we can reach second or minutes timescales, depending on the reactions at hand, far beyond MD scale, no matter if run with MLIPs or other force fields.

Also, we would like to clarify that we do not use an MLIP to predict forces and energies for a known reaction path. Instead, we directly predict reaction barriers using a similar architecture without knowing or finding the minimum energy path. This is notably different from the NequIP example mentioned. We have further clarified these points in the manuscript.

Simulations can also be accelerated by coarse-graining, which both reduces the system’s dimensionality and smooths the energy landscape, thereby extending the

simulation time. For example, reactive MARTINI (Sami and Marrink, JCTC 2023) has been used to model reactions using a coarse-grained model. Is KIMMDY faster with a traditional force field than with a reactive coarse-grained potential, such as reactive MARTINI? If technically feasible with a reasonable effort, the authors should directly compare the two, at least on a limited scale. If it is a considerable feat, it is reasonable to settle for contextualizing coarse-grained reactive potentials in the Discussion. Of course, for coarse-graining potentials, this is also a matter of decreased resolution (but that does not necessarily matter if the objective is inference of reaction features).

In principle, KIMMDY can also be used in combination with coarse-grained or ML potentials, dramatically speeding up the MD ensemble generation step. The authors briefly mention the “near-future plugging in” of ML potentials in Supp, but we consider this a point worth raising in the Discussion, for both coarse-grained and ML potentials (while considering the differences in resolution).

This is a very good point that we are in fact currently working on. We expanded the Discussion and now discuss how KIMMDY can be (quite straightforwardly) extended to coarse-grained models such as MARTINI. What is needed, though, is a robust rate estimator still, which again will require either a good heuristic model or ML based on the atomistic configuration (which any DFT barrier calculation is based on).

Markov State Models have been recently used to describe reaction dynamics (<https://www.pnas.org/doi/10.1073/pnas.2406356121>). At least conceptually, there is some analogy between that work and the proposed methodology (without taking away from its novelty - these works are quite different), as MSMs are used to describe the high-level kinetics. In contrast, short simulation trajectories are used to describe the fine details. The authors may contextualize such alternative approaches in the Discussion.

We discuss this in depth in the revised version: “We recognize parallels to methods leveraging Markov state models (MSMs) to describe reaction kinetics (Li, Yao, and Pan 2025). In contrast to these approaches, KIMMDY does not require computationally expensive ab initio or MLIP MD simulations, but instead allows the use of MM force fields.” (Ab initio methods are only needed to generate training data for machine learned reaction plugins, on systems with drastically lower atom counts.) ” This substantially reduces the computational cost and enables the investigation of larger biomolecular systems, such as the collagen fibrils presented here. Additionally, KIMMDY is readily applicable to systems with similar chemistry without the need for prior computations. However, KIMMDY limits the accessible reaction space to reactions predefined by the reaction plugins, whereas ab initio methods can discover all possible reactions and transition states.”

Optional suggestion: The manuscript acknowledges that the GNN tends to underestimate absolute rates (attributed partly to missing nuclear quantum effects like

tunneling). While the authors have performed a Brier divergence analysis, they could consider validating the barrier’s calibration more rigorously. Specifically, if the model predicts a high probability of a reaction pathway, does this align with ground-truth frequency? A calibration plot comparing predicted vs. observed barrier distributions would be valuable. For example, in the n-alkyl radical benchmark (Fig. 2c, Fig. A1), the authors could compare the model-predicted barrier (or rate) distributions against the experimentally inferred distributions. For each HAT shift type, does the predicted selection probability match the experimental one within confidence intervals?

Please perform at least a limited sensitivity analysis of the barrier predictions with respect to input noise. Do minor fluctuations in the input geometry (thermal noise) cause significant variances in the predicted barrier ranking? Discuss how robustness (or sensitivity) of the GNN to conformational noise may influence the kMC steps, to ensure these steps are physically meaningful rather than artifacts of model instability.

Thank you for this important point about calibration and robustness. We believe that our chosen n-alkyl example system demonstrates this calibration through comparing the distribution of 1-n HAT with the experimental rate distributions in (now) fig. 3c. Additionally, we compute the Brier divergence as mentioned, which directly assesses the “distance“ between the predicted and experimental distributions in fig. A1b. In the mentioned example we only compare single step HAT reactions due to the availability of experimental rates, therefore we cannot meaningfully create a calibration plot for a full multi-step reaction pathway.

For example, in the n-alkyl radical benchmark (Fig. 2c, Fig. A1), the authors could compare the model-predicted barrier (or rate) distributions against the experimentally inferred distributions.

We believe that this point might have been missed in the paper, as this can be seen in (now) fig. 3c. We added some additional clarifying remarks to the paper, thank you for pointing this out.

Concerning the robustness, we believe that our example of the n-alkyl radical demonstrates this, since we run 20 independent simulations for the hydrogen shift reactions for each n-alkyl radical with very similar obtained mean rates, as well as showing that the selection probability in fig. 3b remains approximately stable across the 20 independent runs of 1-octyl and 1-heptyl in (now) fig. 3b. Thermal noise is inherent to MD as we run it, so the resulting similar reaction rates and stable selection probabilities demonstrate this robustness. We more clearly state this in the revised version.

While the cross-domain generalization of GNNs is briefly demonstrated (the n-alkyl radical HAT benchmark), the manuscript could benefit from a practical guide and an example for users applying KIMMDY to systems that differ significantly from the training data. As a potential user, one needs to understand the operational limits and whether and how KIMMDY can be applied ‘out of the box’ to new chemistries,

or whether strictly localized retraining is always required. We leave details to the discretion of the authors, but some examples are as follows: How can a user verify if the pre-trained GNN is valid for their specific, novel biological system? The authors could explicitly quantify how performance degrades as the chemical environment deviates from that of the training set. Does the model provide a confidence score or an uncertainty metric to warn the user when the reaction environment is too distinct to model accurately?

Thank you for raising the important point of cross-domain generalization. Firstly, we would like to emphasise the KIMMDY-framework is agnostic to the method used to obtain rates. We here used a model trained on Collagen data to obtain barriers, since this was also the main application area. The ML methodology used in this work was first published in (Riedmiller et al. 2024) and contains the details of the method. In short, the spread of the ensemble of multiple models is used to quantify the uncertainty. A user could therefore use this methodology to decide whether a pretrained model is suitable for their system. However, as we discuss in the n-alkyl radical case, we expect best results for models that are trained on relevant environments directly, but we nonetheless show that even without this specification, we obtain an overall correct ranking. We emphasize this point more specifically in the expanded section on ‘limitations’ in the Discussion.

We have only briefly reviewed the code, as we were allocated a short timeframe for the review. However, the code appears to be well-maintained and in good shape. We did not have the time to test it ourselves, though the installation seems quite straightforward.

Thank you for this feedback.

Reviewer 5

References

De Keer, Lies, Karsu I. Kilic, Paul H. M. Van Steenberge, Lode Daelemans, Daniel Kodura, Hendrik Frisch, Karen De Clerck, et al. 2021. “Computational Prediction of the Molecular Configuration of Three-Dimensional Network Polymers.” *Nature Materials* 20 (10): 1422–30. <https://doi.org/10.1038/s41563-021-01040-0>.

- Gilley, Dylan, Vignesh Sathyaseelan, and Brett Savoie. 2025. “Escaping Vibrational Purgatory: Hybrid kMC/MD Algorithms for Atomistic Simulations of Slow Reaction Chemistry.” October 9, 2025. <https://doi.org/10.26434/chemrxiv-2025-7tjbx>.
- Li, Chu, Yuan Yao, and Ding Pan. 2025. “Unveiling Hidden Reaction Kinetics of Carbon Dioxide in Supercritical Aqueous Solutions.” *Proceedings of the National Academy of Sciences* 122 (1): e2406356121. <https://doi.org/10.1073/pnas.2406356121>.
- Riedmiller, Kai, Patrick Reiser, Elizaveta Bobkova, Kiril Maltsev, Ganna Gryn’ova, Pascal Friederich, and Frauke Gräter. 2024. “Substituting Density Functional Theory in Reaction Barrier Calculations for Hydrogen Atom Transfer in Proteins.” <https://doi.org/10.1039/D3SC03922F>.

Response to Referees

Please find below our point-by-point responses. Original reviewer comments are indented and shown in light blue.

Reviewer 1

I have carefully checked the revised manuscript and the reply letter. The work has been sufficiently improved in view of publication.

We thank the reviewer for their valuable feedback and are pleased that the revised manuscript satisfactorily addresses their concerns.

Reviewer 3

I think the present version of the manuscript can be accepted.

Minor comments on the SI file: The title of Table S4 should also go above the table. At Fig. S11 caption "Tableau" should be "Table".

We thank the reviewer for their positive assessment. The title placement of Table S4 has been corrected, and the caption of Fig. S11 (now Fig. S17) has been revised accordingly.

Reviewer 4

ok with current version

We thank the reviewer for their positive evaluation of the revised manuscript.

Reviewer 5
